# Diffusion Trajectory-guided Policy: A Novel Framework for Long-Horizon Robot Manipulation

## Abstract

Recently, Vision-Language Models (VLMs) have made substantial progress in robot imitation learning, benefiting from increased amounts of demonstration data. However, the high cost of data collection remains a significant bottleneck, and the scarcity of demonstrations often result in poor generalization of the imitation policy, especially in long-horizon robotic manipulation tasks. To address these challenges, we propose the **D**iffusion **T**rajectory-guided **P**olicy (**DTP**) framework, which generates task-relevant trajectories through a diffusion model to guide policy learning for long-horizon tasks. Furthermore, we demonstrate that our DTP method offers a useful interface for prompt engineering, providing a novel way to connect robot manipulation skills with interactions involving LLMs or humans. Our approach employs a two-stage training process: initially, we train a generative vision-language model to create diffusion task-relevant trajectories, then refine the imitation policy using these trajectories. We validate that the DTP method achieves substantial performance improvements in extensive experiments on the CALVIN simulation benchmark, starting from scratch without any external pretraining. Our approach outperforms state-of-the-art baselines by an average of 25% in success rate across various settings.

## 1 Introduction

Imitation Learning (IL) demonstrates significant potential in addressing manipulation tasks within real robotic systems, this is evidenced by its ability to acquire diverse behaviors such as preparing coffee (Zhu et al., 2023) and flipping mugs (Chi et al., 2023) through learning from expert demonstrations. However, these demonstrations often fail to encompass every potential robot pose and environment variation, from start to finish of tasks in long-horizon manipulation (Fig. 1(a)). Moreover, unlike tasks in natural language processing (NLP) and computer vision (CV) (He et al., 2022; Achiam et al., 2023; Li et al., 2022), the IL faces significant challenges due to the disparate semantic features between vision, language, and action spaces. Additionally, robot data is often sparse compared to NLP and CV tasks because collecting it requires costly and time-consuming human demonstrations. Therefore, improving the generalization capabilities of imitation learning methods using extremely limited and sparse data, given the constraints and high costs of expert demonstrations, becomes a significant challenge.

To address this challenge, recent research has proposed Vision-Language Action (VLA) models (Brohan et al., 2022; 2023; Ma et al., 2024) to map multi-modal inputs to robot actions by using transformer structures (Vaswani, 2017). For model input, several approaches integrate vision and language to generate a goal image, as seen in methods like Susie (Black et al., 2023) or future videos (Du et al., 2023; 2024), which are pretrained on large-scale video dataset from internet. The RT-trajectory (Gu et al., 2024) uses coarse trajectory sketches as modality instead of language, while the RT-H (Belkhale et al., 2024) involves breaking down complex language instructions into simpler, hierarchical commands. For example, instruction as "Close the pistachio jar" can be decomposed step by step into actions like "rotate arm right", "move arm forward", etc., thereby facilitating robot action generation. These methods share a common goal of reducing the feature disparity between the language and action spaces. This includes approaches such as transferring complex language to a goal image, which then generates the action, replacing language instructions with coarse trajectory

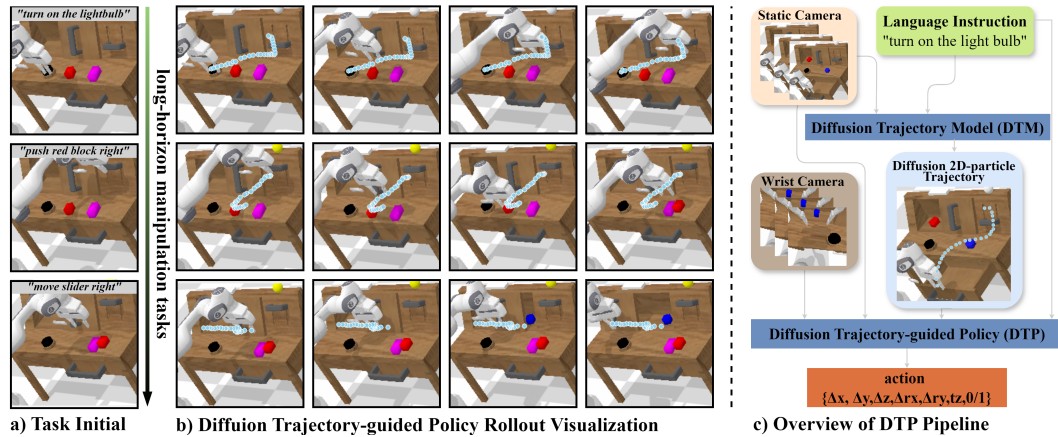

Figure 1: **Overview.** The left side presents a task instruction with the initial task observation, allowing our Diffusion Trajectory Model to predict the complete future 2D-particle trajectories. The right side illustrates the Diffusion Trajectory-guided pipeline, showcasing how these predicted trajectories guide the manipulation policy for effective task execution.

sketches that are more intuitive for the action space, or simplifying language instructions into directional commands that are easier to map to actions, thereby facilitating more effective task execution. For model output, the Diffusion Policy (Chi et al., 2023) offers a unique perspective by defining action outputs as generative tasks, similar to image generation (Ho et al., 2022). This novel insight presents a promising method to address the generalization challenges in imitation learning policies.

In this paper, we introduce a novel diffusion-based paradigm designed to reduce the feature disparity between the vision-language input and action spaces. By using vision-language input to generate task-relevant 2D trajectories, which are then mapped to the action space, our approach enhances performance in long-horizon robotic manipulation tasks. Unlike robots, which often rely on precise instructions, humans use high-level visualization, such as imagined task-relevant trajectories, to intuitively guide their actions. This visualization aids in adapting to changing conditions and refining our movements in real-time. Similarly, when instructing a robot using language, it should be feasible to envision a task-relevant trajectory to guide the robot's future actions based on current observations. To facilitate this process, We introduce the **D**iffusion **T**rajectory-guided **P**olicy (DTP), which consists of two stages: the **D**iffusion **T**rajectory **M**odel (DTM) learning stage and the vision-language action policy learning stage. The first stage involves generating a task-relevant trajectory based on a diffusion model. In the second stage, this diffusion trajectory serves as a guiding framework for the robot's manipulation policy, enabling the robot to perform tasks with better data efficiency and improved generalization. We validated our method through extensive experiments on the CALVIN simulation benchmark (Mees et al., 2022b), where it outperformed state-of-the-art baselines by an average success rate of 25% across various settings. Additionally, Our approach is computationally cost-effective requiring only consumer-grade GPUs.

The main contributions of the paper include:

1. We propose the DTP, a novel imitation learning framework that utilizes a diffusion trajectory model to guide policy learning for long-horizon robot manipulation tasks.

2. Instead of relying on costly large-scale pretraining methods, we leverage robot video data to pretrain a generative vision-language diffusion model. This approach enhances imitation policy training efficiency by fully utilizing available robot data. Furthermore, our method can be combined with large-scale pretraining methods, serving as a simple and effective plugin to enhance performance.

3. We validate the effectiveness of our method through extensive simulated experiments, assessing DTP's performance across diverse settings. Our method achieves a 25% higher success rate compared to state-of-the-art baseline method.

## 2 RELATED WORK

**Language-conditioned Visual Manipulation Policy Control.** Language-conditioned visual manipulation has made significant progress due to advancements in large language models (LLMs) and vision-language models (VLMs). By using task planners like GPT-4 Achiam et al. (2023) or Palm-E Driess et al. (2023), it is possible to break down complex embodied tasks into simpler, naturally articulated instructions. If robotic manipulation could be fully controlled through natural language instructions, akin to human execution, it could usher in a new generation of intelligent embodied agents. Recently, several innovative methods have been developed in this domain. RT-1 Brohan et al. (2022) pioneered the end-to-end generation of actions for robotic tasks. RT-2 Brohan et al. (2023) explores the capabilities of LLMs for Vision-Language-Action (VLA) tasks by leveraging large-scale internet data. RoboFlamingo Li et al. (2024a) follows a similar motivation as RT-2, focusing on the utilization of extensive datasets. RT-X prioritizes the accumulation of additional robotic demonstration data to refine training and establish scaling laws in robotic tasks. The Diffusion Policy Chi et al. (2023) addresses the prediction of robot actions using a denoising model. Lastly, Octo Octo Model Team et al. (2024) serves as a framework for integrating the aforementioned contributions into a unified system, further advancing the filed of language-conditioned visual manipulation.

**Policy Conditioning Representations.** Due to the high-dimensional semantic information contained in language, using video prediction as a pre-training method Du et al. (2024); Escontrela et al. (2024) yields reasonable results. In these approaches, a video prediction model generates future subgoals, which the policy then learns to achieve. Similarly, the goal image generation method Black et al. (2023) utilizes images of subgoals instead of predicting entire video sequences for policy learning. However, both video prediction and goal image generation models often produce hallucinations and unrealistic physical movements. Additionally, these pre-training models demand significant computational resources, posing challenges particularly during inference. RT-trajectory Gu et al. (2024) and ATM Wen et al. (2023) offer innovative perspectives on generating coarse or particle trajectories, which have proven effective and intuitive. Inspired by these approaches, our method introduces unique adaptations. Unlike RT-trajectory, which generates relatively coarse trajectories through image generation or sketch, our method does not completely replace language instructions with coarse trajectories. Instead, we produce high-quality trajectories that can be directly used for end-to-end model inference. Additionally, we use particle trajectories rather than linear trajectories, allowing for more precise and flexible task execution. In contrast to ATM, we model the entire task process using a single key point representing the end-gripper's position in RGB. To unify the concept of 2D points or waypoints in the RGB domain, We refer to the series of key points from the start to the end of a task as 2D-particle trajectories.(Fig. 1(b)). Our method functions similarly to video prediction, serving as a plugin to enhance policy learning. Furthermore, extensive experiments confirm that our approach does not conflict with video pre-training methods. We perform our method using the GR-1 framework Wu et al. (2024), which incorporates a causal transformer Radford (2018) and video pre-training method. With the GR-1 baseline, integrating particle trajectories as an additional input proved straightforward, and our evaluations confirmed that our method does not conflict with existing video pre-training approaches.

**Diffusion Model for Generation.** Diffusion models in robotics are primarily utilized in two areas. Firstly, as previously discussed, they are used for generating future imagery in both video and goal image generation tasks. Secondly, diffusion models are applied to visuomotor policy development, as detailed in recent studies Chi et al. (2023); Reuss et al. (2024); Octo Model Team et al. (2024). These applications highlight the versatility of diffusion models in enhancing robotic functionalities. Unlike other methods, our approach does not use diffusion models to directly generate the final policy. Given the high-dimensional semantic richness of language, we propose utilizing diffusion models to create a 2D-particle trajectory. This trajectory represents future end-gripper movements planing in the RGB domain. We believe that such diffusion trajectories, which contain more detailed information, simplify the policy learning process and enhance its effectiveness.

## 3 METHOD

Our goal is to create a policy that enables robots to handle long-horizon manipulation tasks by interpreting vision and language inputs. We simplify the VLA task using two distinct phases

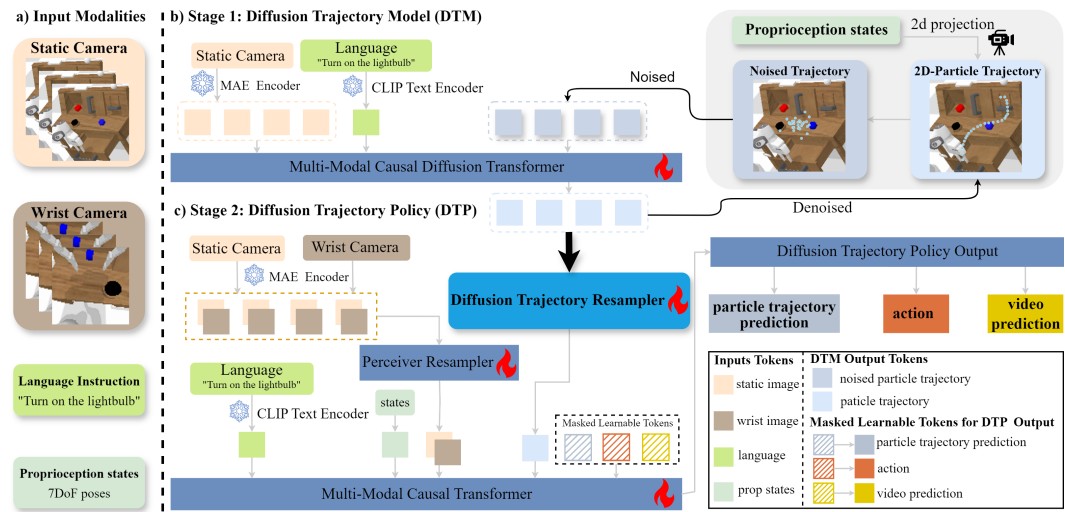

Figure 2: **Network Architecture** for learning language-conditioned policies. a) Shows the input modalities, including vision, language, and proprioception. b) Describes the Diffusion Trajectory Model, detailing how vision and language inputs generate diffusion particle trajectories. c) Explains how these trajectories guide the training of robot policies, focusing on the learning of the Diffusion Trajectory Policy. Masked learnable tokens represent the particle trajectory prediction token, action token, and video prediction token, respectively. These masked tokens serve as the output of the policy.

(Fig. 2(b)(c)): a Diffusion Trajectory Model (DTM) learning phase and a Diffusion Trajectory Policy (DTP) learning phase. Initially we generate the diffusion 2D-particle trajectory for the complete task. Subsequently, in the second stage, we utilize these 2D-particle trajectories to guide the learning of the manipulation policy.

## 3.1 PROBLEM FORMULATION

**Multi-Task Visual Robot Manipulation.** We consider the problem of learning a language-conditioned policy $\pi_\theta$ that take advantage of language instruction $l$, observation $\boldsymbol{o}_t$, robot states $\boldsymbol{s}_t$ and diffusion trajectory $\boldsymbol{p}_{t:T}$ to generate a robot action $\boldsymbol{a}_t$:

$$\pi_\theta(l, \boldsymbol{o}_t, \boldsymbol{s}_t, \boldsymbol{p}_{t:T}) \rightarrow \boldsymbol{a}_t \qquad (1)$$

The robot receives language instructions detailing its objectives, such as "turn on the light bulb". The observation sequence, $\boldsymbol{o}_{t-h:t}$, captures the environment's data from the previous $h$ time steps. The state sequence, $\boldsymbol{s}_{t-h:t}$, records the robot's configurations, including the pose of the end-effector and the status of the gripper. The diffusion trajectory, $\boldsymbol{p}_{t:T}$, predicts the future movement of the end-gripper from time $t$ to the task's completion at time $T$. Our dataset, $\mathbb{D}$, comprises $n$ expert trajectories across $m$ different tasks, denoted as $\mathbb{D}_m = \{\tau_i\}_{i=1}^n$. Each expert trajectory $\tau$ includes a language instruction along with a sequence of observation images, robot states, and actions: $\tau = \{\{l, \boldsymbol{o}_1, \boldsymbol{s}_1, \boldsymbol{a}_1\} ..., \{l, \boldsymbol{o}_T, \boldsymbol{s}_T, \boldsymbol{a}_T\}\}$.

## 3.2 FRAMEWORK

We introduce the Diffusion Trajectory-guided Policy, as illustrated in Fig. 2. DTP operates within a two-stage framework. In the first stage, our primary focus is on generating the diffusion trajectory $\boldsymbol{p}_{t:T}$ which outlines the motion trends essential for completing the task, as observed from a static perspective camera (Fig. 2(b) right part). This 2D-particle trajectory serves as the guidance for subsequent policy learning using a baseline model GR-1. GR-1 is a causal transformer Radford (2018) designed to handle diverse modalities, processing inputs to predict future images and robotic actions with learnable observation and action query tokens respectively. It integrates CLIP (Radford et al., 2021) as the language encoder for processing language instructions $l$, with frozen parameters,

and employs a MAE (He et al., 2022) for the vision encoder $\boldsymbol{o}_{t-h:t}$, also with frozen parameters. The vision tokens are then processed with a perceiver resampler (Jaegle et al., 2021) to reduce their number. Additionally, it incorporates the robot's state $\boldsymbol{s}_{t-h:t}$ in world coordinates, as part of its input. All input modalities are shown in Fig. 2(a). For more detailed information, refer to GR-1 Wu et al. (2024). The reason for incorporating this baseline into our framework is detailed in Section 4.3. Our approach is divided into two main sections. Initially, we detail the process of learning a diffusion trajectory model from the dataset $\mathbb{D}$ in Section 3.3. Subsequently, in Section 3.4, we illustrate how the diffusion trajectory can guide the policy learning for long-horizon robot tasks.

## 3.3 DIFFUSION TRAJECTORY MODEL

In the first stage (Fig. 2(b)), we focus on generating diffusion trajectory that maps out the motion trends required for task completion, as viewed from a static perspective camera. To achieve this, we employ a model $\boldsymbol{M}_d$ to transform language instructions $l$ and initial visual observations $\boldsymbol{o}_t$ into a sequence of diffusion 2D-particle trajectories $\boldsymbol{p}_{t:T}$. These points indicate the anticipated movements for the remainder of the task:

$$\boldsymbol{M}_d(l, \boldsymbol{o}_t) \rightarrow \boldsymbol{p}_{t:T} \tag{2}$$

### 3.3.1 DATA PREPARATION

According to Eq. 2, our input consists of observations $\boldsymbol{o}_t$ and language instructions $l$, as provided by the CALVIN Benchmark (Mees et al., 2022b). For outputs, our aim is to determine the future 2D-particle trajectory $\boldsymbol{p}_{t:T}$ of the end effector gripper for finishing the task. Recent advancements in video tracking work make it easy to monitor the end effector gripper (Yang et al., 2023). For enhanced convenience and precision, we achieve this by mapping the world coordinates $(x_w, y_w, z_w)$ to pixel-level positions $(x_c, y_c)$ according to camera's intrinsic and extrinsic parameters in the static camera frame, as shown in (Fig. 2(b)) right part. In the first stage of training, our data format is structured as $\mathbb{D}_{\text{trajectory}} = \{l, \boldsymbol{o}_t, \boldsymbol{p}_{t:T}\}$, facilitating straightforward acquisition of the sequence $\boldsymbol{p}_{t:T}$, thereby simplifying the process of training our model to accurately predict end effector positions.

### 3.3.2 TRAINING OBJECTIVE

Denoising Diffusion Probabilistic Models (DDPMs) (Ho et al., 2020) constitute a class of generative models that function operates by predicting and subsequently removing noise during the generation process. In our approach, we utilize a causal diffusion decoding structure (Chi et al., 2023) to generate diffusion 2D-particle trajectories $\boldsymbol{p}_{t:T}$. Specifically, we initiate the generation process by sampling a Gaussian noise vector $x^K \sim \mathcal{N}(0, I)$ and proceed through $K$ denoising steps using a learned denoising network $\epsilon_\theta(x^k, k)$ where $x^k$ represents the diffusion trajectory noised over $K$ steps. This network iteratively predicts and removes noise $K$ times, ultimately resulting in the output $x^0$, which denotes the complete removal of noise. The process is governed by the equation below, where $\alpha$, $\gamma$, and $\sigma$ are parameters that define the noise schedule:

$$x^{k-1} = \alpha(x^k - \gamma\epsilon_\theta(x^k, k)) + \mathcal{N}(0, \sigma^2 I) \tag{3}$$

Eq. 3, illustrates the functioning of the basic diffusion model. For our application, we adapt this model to generate diffusion trajectories $\boldsymbol{p}_{t:T}$ based on conditioned inputs: the observation $\boldsymbol{o}_t$ and language instruction $l$. We modify equation to incorporate these inputs, transforming it as follows:

$$\boldsymbol{p}_{t:T}^{k-1} = \alpha(\boldsymbol{p}_{t:T}^k - \gamma\epsilon_\theta(\boldsymbol{o}_t, l, \boldsymbol{p}_{t:T}^k, k)) + \mathcal{N}(0, \sigma^2 I) \tag{4}$$

During the training process, the loss is calculated as follows, where $\epsilon_k$ represents noise sampled randomly:

$$\mathcal{L}_{DTM} = \text{MSE}(\epsilon_k, \epsilon_\theta(\boldsymbol{o}_t, l, \boldsymbol{p}_{t:T} + \epsilon_k, k)) \tag{5}$$

This transformation integrates our specific inputs into the diffusion process, enabling the tailored generation of diffusion trajectory in alignment with both the observed data and the provided linguistic directives. This training loss ensures that diffusion 2D-particle trajectories are accurately generated by systematically reducing noise, thereby enhancing the clarity and precision of the final trajectory predictions. For more detailed information on the DTM algorithm pipeline, refer to App. A.1. Training hyperparameters are listed in Tab. 3. The visualization of DTM is provided in Appendix A.4.

### 3.4 DIFFUSION TRAJECTORY-GUIDED POLICY

In the second stage, we focus on illustrating how the diffusion trajectory guides the robot manipulation policy (Fig. 2(c)). As previously outlined in our problem formulation, we define our task as a language-conditioned visual robot manipulation task. We base our Diffusion Trajectory-guided Policy on the GR-1 (Wu et al., 2024) baseline model and incorporate our diffusion trajectory $p_{t:T}$ as an additional input, as specified in Eq. 1.

**Baseline Policy Input.** This consists of language and image inputs, as detailed in the Sec. 3.2 and shown in the left side of Fig. 2(c). To clearly demonstrate our method's performance, we maintain the same configuration as GR-1.

**Diffusion Trajectory as Extra Policy Input.** Importantly, for the diffusion trajectory, we do not rely on the inference results from the first training stage. Instead, we use the labeled data from this stage as the diffusion trajectory. This approach enhances precision in training and conserves computational resources, by using the labels directly. The simplest training approach is to inject the diffusion particle trajectory directly into the causal baseline. However, our fixed set of 2D particle trajectories $p_{t:T}$ can lead to computational intensity during training due to the high number of tokens. Inspired by the perceiver resampler Jaegle et al. (2021), we designed a diffusion trajectory resampler module to reduce the number of trajectory tokens, as shown in Fig. 2(b) and (c).

**Diffusion Trajectory as Policy Training.** During the policy learning phase (Fig. 2(c)), we generate future particle trajectories to supervise the diffusion trajectory resampler module and the baseline attention module. Our policy framework also employs a causal transformer architecture, similar to the baseline model GR-1 setting, where future particle trajectory tokens are generated prior to action tokens, This sequencing ensures that the particle trajectory tokens effectively guide the formation of action tokens, optimizing the action prediction process in a contextually relevant manner. Additionally, we retain the output of video prediction, maintaining the same setting as GR-1. This consistency in output makes it easier to conduct ablation studies, as we can directly compare our approach to the original GR-1 model.

$$\mathcal{L}_{DTP} = \mathcal{L}_{trajectory} + \mathcal{L}_{action} + \mathcal{L}_{video} \tag{6}$$

Furthermore, to demonstrate the effectiveness and superiority of our method in the ablation study, we split the GR-1 baseline into two versions: one that is fully pretrained on the video dataset and another that only uses the GR-1 structure without any pretraining. We will discuss these two baseline configurations in Sec. 4. More details about the inference process of the DTP are provided in App. 2. Training hyperparameters are listed in Tab. 3.

## 4 EXPERIMENT

In this section, we evaluate the performance of Diffusion Trajectory Policy on the CALVIN benchmark (Mees et al., 2022b). The experiments aim to answer the following questions:

1. How does DTP perform in long-horizon manipulation tasks compared against state-of-the-art baseline methods?

2. Does the DTP enhance the baseline model's performance in long-horizon manipulation tasks, and does it improve the efficiency of imitation policy training by utilizing only the robot data provided?

3. Can DTP achieve data efficiency in solving language-conditioned multi-task problems?

4. What emergent capabilities are enabled by DTP?

### 4.1 CALVIN BENCHMARK AND BASELINE

CALVIN (Mees et al., 2022b) is a comprehensive benchmark designed for evaluating language-conditioned policies in long-horizon robot manipulation tasks. It comprises four distinct yet similar environments (A,B,C, and D) which vary in desk shades and item layouts, as shown in Fig. 3. This benchmark includes 34 manipulation tasks with unconstrained language instructions. Each

Table 1: Summary of Experiments: This table details the performance of all baseline methods in sequentially completing 1, 2, 3, 4, and 5 tasks in a row. The average length, shown in the last column and calculated by averaging the number of completed tasks in a series of 5 across all evaluated sequences, illustrates the models' long-horizon capabilities. 10%ABCD→D indicates that only 10% of the training data is used.

| Method | Experiment | Tasks completed in a row | | | | | Avg. Len. |
|---|---|---|---|---|---|---|---|
| | | 1 | 2 | 3 | 4 | 5 | |
| HULC | D→D | 0.827 | 0.649 | 0.504 | 0.385 | 0.283 | 2.64 |
| GR-1 | D→D | 0.822 | 0.653 | 0.491 | 0.386 | 0.294 | 2.65 |
| MT-ACT | D→D | 0.884 | 0.722 | 0.572 | 0.449 | 0.353 | 3.03 |
| HULC++ | D→D | 0.930 | 0.790 | 0.640 | 0.520 | 0.400 | 3.30 |
| DTP(Ours) | D→D | 0.924 | 0.819 | 0.702 | 0.603 | 0.509 | 3.55 |
| HULC | ABC→D | 0.418 | 0.165 | 0.057 | 0.019 | 0.011 | 0.67 |
| RT-1 | ABC→D | 0.533 | 0.222 | 0.094 | 0.038 | 0.013 | 0.90 |
| RoboFlamingo | ABC→D | 0.824 | 0.619 | 0.466 | 0.380 | 0.260 | 2.69 |
| GR-1 | ABC→D | 0.854 | 0.712 | 0.596 | 0.497 | 0.401 | 3.06 |
| 3D Diffuser Actor | ABC→D | 0.922 | 0.787 | 0.639 | 0.512 | 0.412 | 3.27 |
| DTP(Ours) | ABC→D | 0.890 | 0.773 | 0.679 | 0.592 | 0.497 | 3.43 |
| RT-1 | 10%ABCD→D | 0.249 | 0.069 | 0.015 | 0.006 | 0.000 | 0.34 |
| HULC | 10%ABCD→D | 0.668 | 0.295 | 0.103 | 0.032 | 0.013 | 1.11 |
| GR-1 | 10%ABCD→D | 0.778 | 0.533 | 0.332 | 0.218 | 0.139 | 2.00 |
| DTP(Ours) | 10%ABCD→D | 0.813 | 0.623 | 0.477 | 0.364 | 0.275 | 2.55 |

environment features a Franka Emika Panda robot equipped with a parallel-jaw gripper, and a desk that includes a sliding door, a drawable drawer, color-varied blocks, an LED, and a light bulb, all of which can be interacted with or manipulated.

**Experiment Setup.** we train DTP to predict relative action in $xyz$ positions and Euler angles for arm movements, alongside binary actions for the gripper. The training dataset comprises over 20,000 expert trajectories from four scenes (A,B,C, and D), each paired with language instruction labels. Notably, the CALVIN dataset includes 24 hours of tele-operated, undirected play data. To simulate real-world conditions, only 1% of this data is labeled with crowd-sourced language instructions, forming the basis for our training. All methodologies are assessed using the long-horizon benchmark, featuring 1,000 unique sequences of instruction chains articulated in natural language. Each sequence requires the robot to sequentially complete five tasks. During rollouts, the agent receives a reward of 1 for each successfully completed instruction, with a potential total of 5 per rollout. **Baseline.** We compare our proposed policy against the following state-of-the-art language-conditioned multi-task policies on CALVIN: **MT-ACT** (Bharadhwaj et al., 2024) is a multitask transformer-based policy with predicts action chunk instead of single actions. **HULC** (Mees et al., 2022a) is a hierarchical approach which predicts latent features of subgoals based on language instructions and observation. These subgoals are then fed into lower-level policies to generate robot action. **RT-1** (Brohan et al., 2022) represents the first approach that utilizes convolutional layers and transformers to generate actions in an end-to-end manner, integrating both language and observational inputs. It demonstrates the feasibility of an end-to-end vision-language-action framework in a structured method approach. **RoboFlamingo** (Li et al., 2024b) is a fine-tuned Vision-Language Foundation model with 3 billion parameters. It has an additional recurrent policy head specifically designed for action prediction. Originally pretrained on a vast, internet-scale dataset of images and text, it has been further fine-tuned specifically for the CALVIN benchmark to enhance its performance in robot manipulation tasks. **GR-1** (Wu et al., 2024) leverages pretraining on the Ego4D dataset, which contains massive-scale human-object interactions captured through web videos. With extensive pre-training on large-scale video datasets, GR-1 effectively enhances learning in visual robot manipulation tasks. **3D Diffuser Actor** (Ke et al., 2024) integrates 3D scene representations with diffusion objectives to learn robot policies from demonstrations. It includes a policy equipped with a 3D denoising transformer, which fuses information from the 3D visual scene, language instructions, and proprioceptive data to predict the noise in noised 3D robot pose trajectories. This approach facilitates a comprehensive understanding and execution of complex manipulative tasks.

Figure 3: The top four environments correspond to the CALVIN ABCD settings, differing mainly in the positions of the sliding door, LED, bulb, light switch, button, and desk shades. The bottom section shows a sequence of five long-horizon tasks, each guided by a specific instruction.

## 4.2 COMPARISONS WITH STATE-OF-THE-ART METHODS

**Primary Imitation Performance.** This experiment is conducted in the D→D setting, utilizing about 5,000 expert demonstrations for training. The training process takes approximately 1.5 days on 8 NVIDIA 24G RTX 3090 GPUs. This setting clearly demonstrates the effectiveness and time-efficiency of our method. As shown in Tab. 1, DTP significantly outperforms all baseline methods across all metrics in the context of long-horizon tasks. Specifically, DTP increases the success rate for Task 5 from 0.400 to 0.509 and raises the average successful sequence length from 3.30 to **3.55**. Notably, compared to GR-1, our baseline model, DTP enhances performance across all metrics, with the average sequence length increasing by 33.9%. These results indicate that DTP demonstrates superior performance in long-horizon tasks, particularly as the task length increases. Additionally, we validate that the diffusion trajectory in our DTP framework effectively guide the completion of language-conditioned multi-tasks.

**Unseen Scene Results.** This experiment is conducted in the ABC→D setting, which is particularly challenging: models are trained using data from environments A, B, and C and then tested in environment D, an unseen setting during the training phase. The training process takes approximately 5 days on 8 NVIDIA 24GB RTX 3090 GPUs. This experimental setting tests the model's generalization capabilities in a new environment. The results are presented in Tab. 1. When comparing the GR-1 framework, our baseline, with our DTP method, there is an increase in the average task completion length from 3.06 to **3.43**. Additionally, the success rate for completing Task 5 increased to 0.497, the highest recorded value. Notably, even though our method does not use depth modality for training, it outperformed the 3D Diffuser Actor in these tests. This underscores a critical insight: DTP can effectively guide policy learning for long-horizon robot tasks in challenging settings.

**Data Efficiency.** Robot data is more costly and scarce compared to vision-language data. To evaluate data efficiency, we trained using only 10% of the full dataset in the ABCD→D setting, randomly selecting around 2,000 expert demonstrations from over 20,000 episodes. With 34 task types, we collected about 60 demonstrations per task, which is sufficient for effective training in real robot environments. Training took approximately 1 day on 8 NVIDIA 24GB RTX 3090 GPUs. We evaluated across different scenes to simulate diverse real-world environments, which also aids manipulation tasks. The results are shown in Tab. 1. While performance declines for all methods compared to training on the full dataset. , the best baseline method, GR-1, achieves a success rate of 0.778 with an average length of 2.00. DTP shows clear benefits for long-horizon tasks; as task numbers increase, the success rate rises, and the average length reaches **2.55**, outperforming other methods. This highlights DTP's data efficiency. Imitation learning helps the model learn positional preferences, which are essential in long-horizon tasks. When the robot starts from an unseen position, task failures are more likely. However, DTP guides the robot arm with a diffusion trajectory, providing the correct path. Thus, even with fewer demonstrations, DTP quickly acquires the necessary skills.

## 4.3 ABLATION STUDIES

In this section, we conduct ablation studies to evaluate how the diffusion trajectory improves policy learning in visual robot manipulation tasks. The diffusion trajectory, our key contribution, significantly boosts the efficiency of imitation policy training by fully utilizing available robot data. Furthermore, when integrated with large-scale pretraining baseline methods, our approach serves as a straightforward and effective enhancement to performance. To measure the effectiveness of our method, we contrast it with two fundamental baselines. The first baseline employs the GR-1 frame-

Table 2: Ablation Studies: Pre-Training indicates whether we use only the baseline model structure or the baseline pre-trained on the Ego4D dataset. In our ablation studies, we established these two baselines to evaluate the effectiveness and compatibility of our DTM method with other approaches. 10%ABCD→D indicates that only 10% of the training data is used. 100%✓indicates DTM trained on full ABCD→D.

| Pre-Training | DTP (Ours) | Data | 1 | 2 | 3 | 4 | 5 | Avg. Len. |
|---|---|---|---|---|---|---|---|---|
| × | × | ABC→D | 0.815 | 0.651 | 0.498 | 0.392 | 0.297 | 2.65 |
| × | ✓ | ABC→D | 0.869 | 0.751 | 0.636 | 0.549 | 0.465 | 3.27 |
| × | × | 10%ABCD→D | 0.698 | 0.415 | 0.223 | 0.133 | 0.052 | 1.52 |
| × | ✓ | 10%ABCD→D | 0.742 | 0.511 | 0.372 | 0.269 | 0.188 | 2.08 |
| ✓ | × | ABC→D | 0.854 | 0.712 | 0.596 | 0.497 | 0.401 | 3.06 |
| ✓ | ✓ | ABC→D | 0.890 | 0.773 | 0.679 | 0.592 | 0.497 | 3.43 |
| ✓ | × | 10%ABCD→D | 0.778 | 0.533 | 0.332 | 0.218 | 0.139 | 2.00 |
| ✓ | ✓ | 10%ABCD→D | 0.813 | 0.623 | 0.477 | 0.364 | 0.275 | 2.55 |
| ✓ | 100%✓ | 10%ABCD→D | 0.822 | 0.643 | 0.526 | 0.416 | 0.302 | 2.71 |

work (Sec. 3.2) without video pretraining, while the second utilizes large-scale video pretraining with the Ego4D dataset (Grauman et al., 2022), also based on GR-1 framework. Two baselines are established to verify the efficacy and compatibility of our method with other approaches. The more detail for specific task successful rate improvement in show in Fig. 5.

**Diffusion Trajectory Policy Scratch.** First, we evaluate our method in the ABC→D and 10% ABCD→D settings, as shown in the upper part of Tab. 2. The results demonstrate that our diffusion trajectory method significantly enhances performance even without any pretraining. Specifically, our method not only excels in sequentially completed tasks but also shows notable gains in the average task completion length for long-horizon tasks increase of 23.4%. Notably, the success rate for the task 5, which is indicative of the overall long-horizon success, has risen by 56.6%. When compared with the 3D Diffuser Actor, as shown in Tab. 1, despite not utilizing depth modality, our approach matches the SOTA average task completion length of 3.27 on the current leaderboard. This highlights our method's efficiency and capability in handling complex robot manipulation tasks without the need for depth data.

**Diffusion Trajectory Policy with Video Pretrain.** As illustrated in the bottom part of Tab. 2, the variants utilizing our diffusion trajectory effectively serve as a plugin, boosting baseline model performance to state-of-the-art levels. We evaluated our method under both the ABC→D and 10% ABCD→D settings, and the results consistently show improvements over the traditional scratch training method. This clearly indicates that our approach complements and significantly enhances baseline performance across various benchmarks. Additionally, the success rates for each subsequent task show notable increases, with the growth rate rising from 4.2% in the first task to 23.9% in the fifth task. These outcomes further validate that DTP can substantially improve performance in long-horizon manipulation tasks.

**Diffusion Trajectory Model Scaling Law.** The last row highlights the initial training stage of our Diffusion Trajectory Model. Increasing the training data allows the model to generate more accurate points, enhancing the Diffusion Trajectory Policy (DTP). The bottom row demonstrates that even with limited demonstration data for imitation learning, scaling up the training for the diffusion trajectory can significantly improve both the success rate and average task completion length. This experimental setup points to a potential direction: although robot demonstration data is costly to obtain, the data for the DTM is relatively easy to annotate. Individuals only need to sketch a coarse trajectory on an RGB image and associate it with relevant language instructions. This method provides a cost-effective and efficient way to augment data, potentially revolutionizing how we train models for robotic manipulation.

## 4.4 EMERGENT CAPABILITIES

In this section, we discuss the enhancement of the robotic policies through visual prompt engineering, analogous to the use of prompts in LLMs (Wei et al., 2022). We explore strategies to optimize our method for better performance in manipulation tasks. This approach offers a novel methodology for integrating fundamental robotic skills with task planning (Driess et al., 2023).

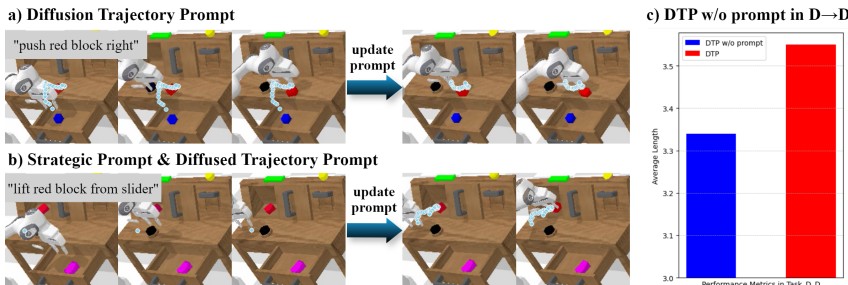

Figure 4: a) The first three frames display the initial diffusion trajectory. The last two frames show the updated diffusion trajectory after object movement to guide the robot. b) Strategic prompts position the robot optimally for task execution in the first three frames and then update the diffusion trajectory to complete tasks. c) These prompts engineering enhance performance in D→D settings.

**Diffusion Trajectory Prompt.** Initially, we generate the diffusion trajectory at the start of each task. However, if the robot's interaction alters the position of an object, such as moving a block without completing the task, it becomes necessary to regenerate the trajectory due to changes in the environment, as shown in Fig. 4(a). The decision to regenerate the trajectory can be made by humans or intelligent systems like LLMs, which can detect changes in the environment's state. In our experiments, we simplify this process with a straightforward strategy: given that manipulation tasks are generally brief, if the duration exceeds a predetermined threshold, we regenerate the diffusion trajectory and restart the task. This approach ensures the trajectory remains relevant and effective throughout the task execution.

We also evaluate prompt engineering in the D→D setting of the CALVIN Benchmark, demonstrating that it enhances performance in long-horizon tasks, with the average task completion length increasing by over 6%. The result is illustrated in Fig. 4(c).

**Strategic Prompt.** A strategic prompt involves drawing particle trajectories using prior knowledge. The entire process is illustrated in Fig. 4(b). More example of strategic prompt by humans or LLMs can be found in App. A.5.

## 5 CONCLUSION AND FUTURE WORK

The limited availability of robot data poses significant challenges in generalizing long-horizon tasks to unseen robot poses and environments. This paper introduces a diffusion trajectory-guided framework that utilizes diffusion trajectories, generated in the RGB domain, to enhance policy learning in long-horizon robot manipulation tasks. This method facilitates the creation of additional training data through data augmentation or manually crafted labels, thereby generating more accuracy diffusion trajectories. Our approach involves two main stages: first, training a diffusion trajectory model to generate task-relevant trajectories; second, using these trajectories to guide the robot's manipulation policy. We validated our method through extensive experiments on the CALVIN simulation benchmark, where it outperformed state-of-the-art baselines by an average success rate of 25% across various settings. Our results confirm that our method not only substantially improves performance using only robot data but also effectively complements and enhances baseline performance across various settings in the CALVIN benchmarks.

In future work, we plan to extend our method to other state-of-the-art policies, as we believe that incorporating diffusion trajectories will further enhance their effectiveness. Another potential direction is to obtain the diffusion trajectory label using the camera's intrinsic and extrinsic parameters, which are not fully available from open-source datasets (Padalkar et al., 2023). Recently, Track-Anything (Yang et al., 2023) demonstrated strong capabilities in tracking arbitrary objects. We could adopt this method to generate diffusion trajectory labels. Furthermore, with similar tracking methods, we can pretrain on large-scale video datasets to train our diffusion trajectory tasks, similar to video prediction tasks. Additionally, implementing our framework in real robot environments represents a crucial next step for future research.

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

## A   APPENDIX

### A.1   METHOD DETAIL

The training and inference process for the Diffusion Trajectory Model is outlined in Alg. 1, corresponding Fig. 2(b).

---

**Algorithm 1:** Diffusion Trajectory Model

---

**Input** : Language Instruction $l$
    Current Observation $\boldsymbol{o}_t$
    Random Sampled Gaussian Noise $\epsilon_k$
    Timesteps for denoising $K$
**Output:** Diffused Trajectory $\epsilon_\theta(\boldsymbol{p}_{t:T}|\boldsymbol{o}_t, l, \epsilon_k)$
    $\boldsymbol{p}_{t:T} = \{(x_t, y_t), \dots, (x_T, y_T)\}$
**Training:**
**for** *each batch* **do**
  Sampling Gaussian Noise $\epsilon_k \sim \mathcal{N}(0, I)$
  Diffused Trajectory with Add Noise $\boldsymbol{p}_{t:T} + \epsilon_k$
  Training Objective MSE$(\epsilon_k, \epsilon_\theta(\boldsymbol{o}_t, l, \boldsymbol{p}_{t:T} + \epsilon_k, k))$
**end**
**Inference:**
Sampling Gaussian Noise $\epsilon_k \sim \mathcal{N}(0, I)$
**for** *timesteps = 1 to K* **do**
  Diffused Trajectory noise predict $\epsilon_{k-\text{timesteps}} = \epsilon_\theta(\boldsymbol{o}_t, l, \epsilon_k, k)$
  $\boldsymbol{p}_{t:T} = \epsilon_k - \epsilon_{k-\text{timesteps}}$
  $\epsilon_k = \epsilon_{k-\text{timesteps}}$
**end**
**return** $\boldsymbol{p}_{t:T}$

---

---

**Algorithm 2:** Diffusion Trajectory Policy Inference

---

**Input** : Language Instruction $l$
    Current Observation $\boldsymbol{o}_t$
    $\boldsymbol{p}_{t:T} = \{(x_t, y_t), \ldots, (x_T, y_T)\}$
**Output:** Particle Trajectory Prediction $\boldsymbol{p}_{t:t+a} = \{(x_t, y_t), \ldots, (x_t + a, y_t + a)\}$
    Action $\boldsymbol{a}_t$
    Video Prediction $\boldsymbol{v}_t$
**Inference:**
Sampling Gaussian Noise $\epsilon_k \sim \mathcal{N}(0, I)$
**for** *t = index to T* **do**
    $l, \boldsymbol{o}_t$ = Robot Observation
    **if** *t==0 or diffusion trajectory prompt == true* **then**
        $\boldsymbol{p}_{t:T}$ = DTM $(l, \boldsymbol{o}_t, \epsilon_k)$
    **end**
    $\boldsymbol{p}_{t:t+a}, \boldsymbol{a}_t, \boldsymbol{v}_t$ = DTP$(l, \boldsymbol{o}_t, \boldsymbol{p}_{t:T})$
    Robot Execute($\boldsymbol{a}_t$)
**end**
**return** Done

---

## A.2 TRAINING HYPERPARAMETERS DETAIL

For training Diffusion Trajectory Model and diffusion Trajectory Policy, an overview of the used hyperparameters is given in Tab. 3. As a result, all experiments were successfully conducted using 8 NVIDIA RTX 3090 (24GB) GPUs, with reproducible results achieved within a few days.

Table 3: Training Diffusion Trajectory Model (DTM) and Diffusion Trajectory Policy (DTP) Hyperparameters.

| Hyperparameters | DTM | DTP |
|---|---|---|
| batch size | 576 | 512 |
| learning rate | 1e-4 | 9e-4 |
| Weight Decay | 1e-6 | 1e-4 |
| Diffusion iterations | 100 | – |
| Trainable Parameters | 454M | 188M |
| 2D-Particle Trajectories | 30 | – |
| dropout | 0.1 | 0.1 |
| optimizer | AdamW | AdamW |
| learning rate schedule | cosine decay | cosine decay |
| warmup epochs | 5 | 5 |
| training epochs | 100 | 50 |

## A.3 PERFORMANCE IMPROVEMENT IN SPECIFIC TASKS

We compared our method with the baseline (Wu et al., 2024) using the CALVIN Benchmark (Mees et al., 2022b) 10% ABCD→D setting to analyze performance improvements across specific tasks. Analyzing Fig. 5 the left group labeled "Interact with blocks" indicates that the robot's task is limited to making contact with blocks, without specific instructions for further interaction with the environment, such as `rotate/push/stack blocks`. According to the graph, the success rate in this comparison group decreases. This decline is likely due to the changing positions of the blocks as the robot interacts with them, necessitating prompt engineering updates to adapt to these new configurations effectively. The middle group, labeled "Interact with blocks based environment," shows an increase in the success rate from 63.24% to 74.68%, demonstrating the benefits of our method. The right group, labeled "Interact with Articulated Object," also shows a 5% increase in success rate. The typical language instructions for the latter two groups are `place/lift blocks to slider/drawer/table` and `open/close drawer, turn on/off lightbulb/LED, move slider right/left,` respectively.

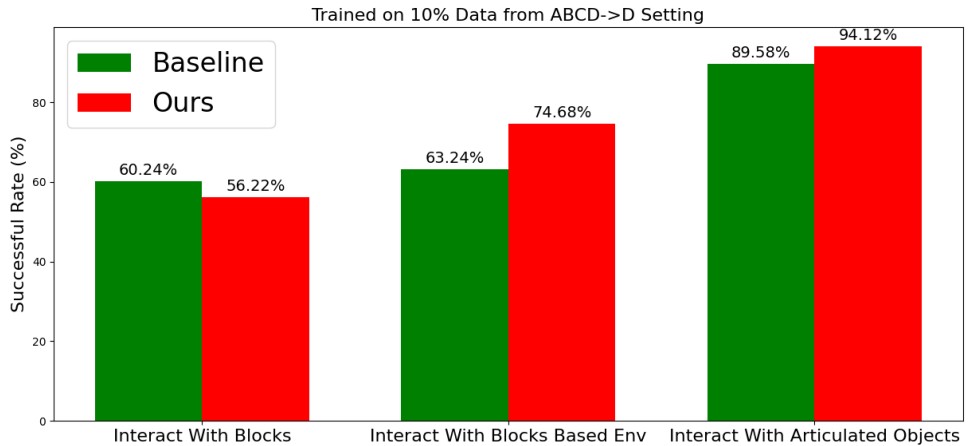

Figure 5: **Performance Improvement in Specific Tasks**. We categorize all manipulation tasks into three types: Interact with Blocks, Interact with Blocks Environment, and Interact with Articulated Objects. Our method shows a slight decrease in performance for "Interact with Blocks," while significantly improving performance in the other two task types.

## A.4 DIFFUSION TRAJECTORY VISUALIZATION

As shown in Fig. 6, we present the overall visualization of the diffusion trajectory generation phase, tested in both the Calvin environment and real-world scenarios. The visualizations demonstrate that the trajectories generated by our diffusion trajectory prediction closely match the ground truth. Even when minor deviations occur, the generated trajectories still align with the robotic arm paths dictated by the language instructions.

## A.5 POTENTIAL PROMPT CAPACITIES WITH HUMANS AND GPT4

**Strategic Prompt.** A strategic prompt involves drawing particle trajectories using prior knowledge. Similar to how LLMs (Achiam et al., 2023) use text prompts, this approach employs 2D coordinates as the format. In long-horizon manipulation tasks, the physical distance between consecutive tasks can be significant, such as moving from the bottom right to the top left. Additionally, the robotic arm may become stuck and fail to move from a certain position. These factors often make it challenging for the robot to assume the correct position and pose, potentially leading to task failure. By implementing strategic prompting, we can guide the robot to an optimal position and pose, significantly enhancing its ability to successfully complete the task. This strategy ensures smoother transitions and more effective task execution. The entire process is illustrated in Fig. 4(b).

The above and main body discusses two types of prompts: diffusion trajectory prompts and strategic prompts.

Diffusion trajectory prompts are used when the position of an object changes, necessitating a re-prompt of the diffusion trajectory to complete tasks successfully. For strategic prompts, we delve deeper in Fig. 7. The left column shows the current observation and the task instruction, which lack detailed positional information. Utilizing strategic prompts, whether provided by humans or large language models (LLMs), significantly enhances the accuracy of placement tasks.

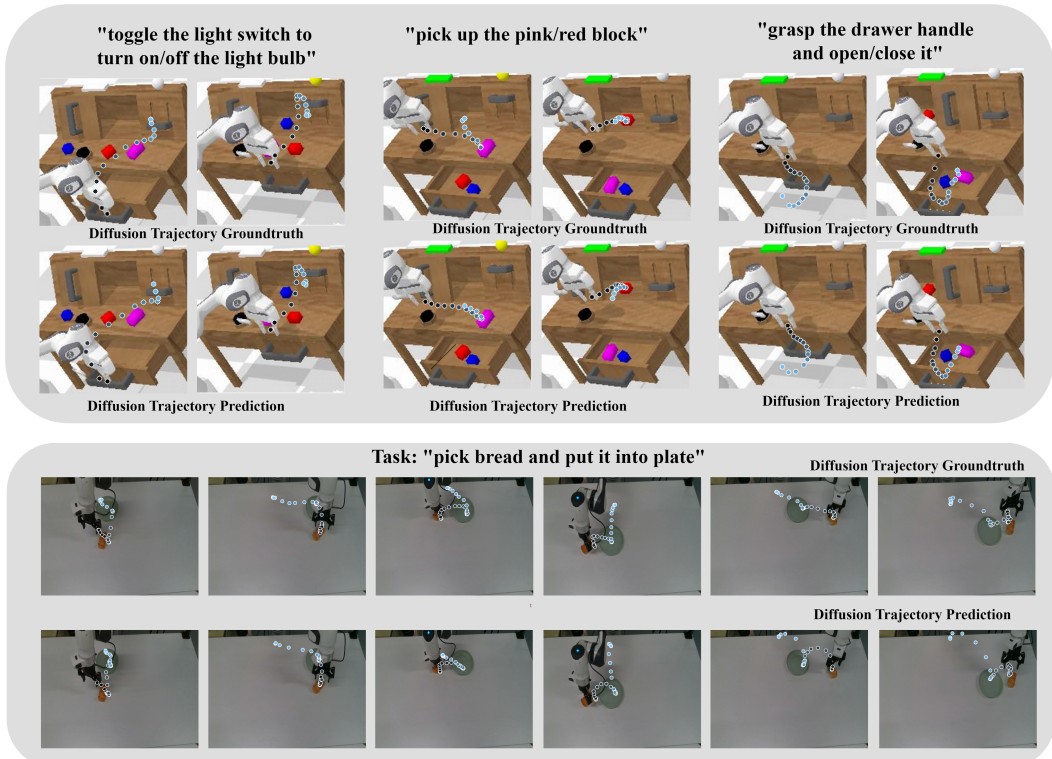

Figure 6: **Diffusion Trajectory Visualization.** The upper section illustrates diffusion trajectory generation in the CALVIN environment, while the lower section depicts trajectory generation in a real-world robotic scenario.

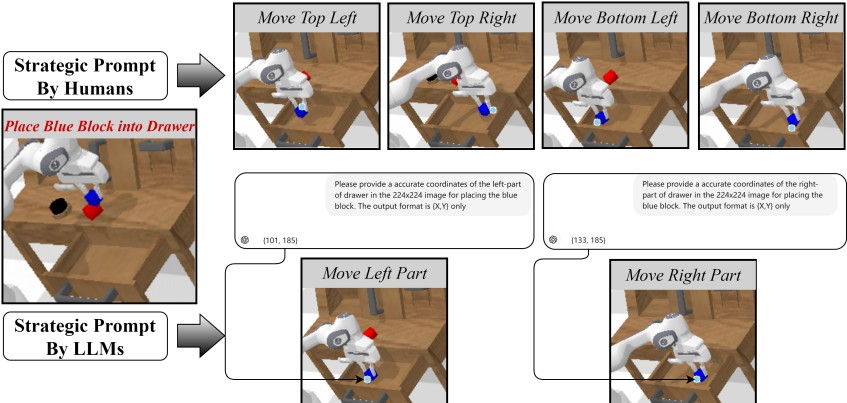

Figure 7: **Prompt Capacities**. The left column represents the current observation and the task instruction, which lacks detailed positional information. Utilizing strategic prompts provided by humans or large language models (LLMs) enhances the ability of the placing task to locate positions with greater accuracy.

