# OpenReview forum: "Diffusion Trajectory-guided Policy: A Novel Framework for Long-Horizon Robot Manipulation"
_ICLR.cc/2025/Conference — Submitted to ICLR 2025_

### Official Review · Reviewer_9oLD · 2024-10-16

**Soundness:** 3
**Presentation:** 3
**Contribution:** 3
**Rating:** 6
**Confidence:** 5

**Summary:**

This paper introduces a diffusion trajectory-guided framework that utilizes diffusion trajectories, generated in the RGB domain, to enhance policy learning in long-horizon robot manipulation tasks.

**Strengths:**

The core idea is compelling, as it utilizes a diffusion-based method to generate trajectories. Additionally, the method's intricate design enables the model to effectively produce particle trajectories.

**Weaknesses:**

-The paper specifically highlights that the generated trajectory can enhance the performance of long-horizon tasks. However, as tasks grow longer, trajectories may overlap. It is unclear how the proposed approach addresses this scenario, and further clarification is needed.

-The paper lacks real-world experiments, making it difficult to assess its practical effectiveness. While real-world execution experiments may be complex, at the very least, visualizations of real-world generated trajectories should be included to provide some insight.

-The introduction mentions that robot data is sparse, yet the diffusion-based trajectory model relies on robot data for training. This inconsistency should be addressed, and claims in the background section should be made more carefully to avoid confusion.

-How effective are the generated trajectories? The authors should design experiments to evaluate this, such as measuring the similarity between generated trajectories and their corresponding ground-truth. Additionally, if the generated trajectory contains noise, it’s important to assess how well the subsequent policy model can handle such inaccuracies.

**Questions:**

Please refer to weakness section

---

> ### Author Response · Authors · 2024-11-25
>
> Dear reviewer:
> Thank you for your detailed review and thoughtful comments. We really appreciate that you think our idea is compelling. We have revised the paper following your suggestions and will address your remaining questions below.
> >The paper specifically highlights that the generated trajectory can enhance the performance of long-horizon tasks. However, as tasks grow longer, trajectories may overlap. It is unclear how the proposed approach addresses this scenario, and further clarification is needed.
>
> Thank you for bringing this up. We agree that as tasks become longer, trajectories may indeed overlap. In our current setup, however, we assume that each individual task within a long-horizon sequence is an atomic-level skill unit that cannot be further divided. Thus, for each atomic-level single task, its trajectory rarely overlaps with others. Additionally, in Fig.4a), we introduce the diffusion trajectory prompt. This feature allows us to generate a new diffusion trajectory mid-sequence if the initial trajectory becomes overly complex, helping to reduce overlap.
>
> >The paper lacks real-world experiments, making it difficult to assess its practical effectiveness. While real-world execution experiments may be complex, at the very least, visualizations of real-world generated trajectories should be included to provide some insight.
>
> Thank you for your kind response. Due to certain practical constraints, we are indeed unable to conduct complex real-world experiments at this time. We greatly appreciate your understanding. You are absolutely right; we should incorporate some real images and conduct predictions on them to enhance the study. In the past few days, we have finally collected some real-world robotic data suitable for training. We conducted training and visualized the results in appendix 4.
>
> >The introduction mentions that robot data is sparse, yet the diffusion-based trajectory model relies on robot data for training. This inconsistency should be addressed, and claims in the background section should be made more carefully to avoid confusion.
>
> Thank you for pointing that out. We sincerely appreciate your helpful suggestion to improve the clarity of our paper. We completely agree; our method aims to make the most of limited robotic data to achieve improved performance. We have revised the introduction section to avoid this inconsistency.
>
> >How effective are the generated trajectories? The authors should design experiments to evaluate this, such as measuring the similarity between generated trajectories and their corresponding ground-truth. Additionally, if the generated trajectory contains noise, it’s important to assess how well the subsequent policy model can handle such inaccuracies.
>
> This is an excellent question, and we were pleasantly surprised by the outstanding results of the generated trajectories. Specifically, we utilized Mean Squared Error (MSE) to evaluate the similarity between the generated trajectories and their corresponding ground truth. The validation loss ultimately converged to below 0.01. Furthermore, we have included visualizations of the first-stage generated trajectories in both real robot and simulation environments, along with their corresponding ground truth, in the appendix 4.

---

### Official Review · Reviewer_mdZa · 2024-10-28

**Soundness:** 3
**Presentation:** 3
**Contribution:** 2
**Rating:** 3
**Confidence:** 3

**Summary:**

This paper introduces a novel imitation learning framework, the Diffusion Trajectory-guided Policy (DTP), designed to address challenges in long-horizon robot manipulation tasks. The core contribution of the paper is the use of diffusion models to generate task-relevant 2D-particle trajectories that guide the learning of a robot’s manipulation policy. The framework operates in two stages: the first stage involves training a diffusion trajectory model (DTM) using language and vision inputs to generate trajectories, and the second stage uses these trajectories to refine the robot's policy. The authors validate their approach on the CALVIN benchmark, demonstrating significant improvements in performance over baseline methods in various long-horizon manipulation tasks.

**Strengths:**

1. The paper presents an innovative use of diffusion models to generate task-relevant trajectories.
2. The method achieves substantial performance gains, with a 25% higher success rate over baselines.

**Weaknesses:**

1. Lack of real-world robot experiments.
2. Lack of novelty. There seems to be no too much takeaway from the designed method, as a lot of works have try to incorporate the diffusion process into long-horizon robotic manipulation.

**Questions:**

See weakness.

---

> ### Author Response · Authors · 2024-11-25
>
> Dear reviewer:
> Thank you for your thoughtful review and constructive comments. We appreciate your recognition that using a diffusion model to generate task-relevant trajectories is a novel approach. We are also grateful for your acknowledgment of the significant improvements our method demonstrates in the experiments. In response to your comments, we would like to address the following points:
>
> >Lack of real-world robot experiments.
>
> Thank you for bringing this up. Due to certain practical constraints, we were unable to conduct complex real-world experiments earlier. However, over the past few days, we successfully collected some real-world robotic data suitable for training diffusion trajectories. The results, including visualizations of the generated 2D trajectories, are now included in Appendix 4.
>
> >Lack of novelty. There seems to be no too much takeaway from the designed method, as a lot of works have try to incorporate the diffusion process into long-horizon robotic manipulation.
>
> Thank you for pointing that out. Indeed, there have been several recent works on diffusion models in manipulation, such as [1,2]. However, these works mainly focus on the output stage, while our approach differs significantly in that we use a diffusion model to generate task-relevant trajectories. This allows the model to learn the policy more effectively. The underlying motivation and approach are therefore fundamentally different from previous works.
> If you have any further questions or concerns, please feel free to point them out. Thank you again for your valuable feedback.
>
> ### References
> [1] http://arxiv.org/abs/2303.04137
>
> [2] http://arxiv.org/abs/2407.05996

---

### Official Review · Reviewer_EuXe · 2024-11-01

**Soundness:** 3
**Presentation:** 2
**Contribution:** 2
**Rating:** 3
**Confidence:** 4

**Summary:**

The authors propose an imitation learning method that additionally conditions on predicted robot gripper trajectories represented as a sequence of 2D key-points. In the first stage, given an initial observation and language instruction, a diffusion model is trained to generate a full sequence of key-points that correspond to the gripper motion that completes the task. In the second stage, a policy is trained to predict actions given the current observation, language instruction, and sequence of gripper key-points. For the policy, the authors fine-tune a VLA model from prior work, GR-1, that was pre-trained on video prediction. At test time the diffusion model is used to generate the sequence of key-points to condition the policy. Experiments on the CALVIN simulated imitation learning benchmark demonstrate that this method outperforms prior methods on long-horizon language-conditioned tasks. The method improves performance with or without the video pre-training from GR-1, though video pre-training does improve the results. Experiments also find that scaling data for only the key-point sequence predictor improves downstream performance.

**Strengths:**

The paper addresses an important problem, long-horizon language-conditioned imitation learning, with a combination of powerful techniques: generative modeling with diffusion and pre-training robot policies on vision-language tasks (i.e using GR-1). Generating intermediate state/action representations for policy conditioning is also an active area of research at the moment, so it's likely the community would find this interesting.

The paper compares to relevant prior work on a well-established baseline, CALVIN, and performs a couple interesting ablations, (1) removing the GR-1 pre-training and (2) limiting data for just the policy. The method's performance on CALVIN is state-of-the-art as of when the paper was submitted.

**Weaknesses:**

**Motivation**

The motivation of the method is not entirely clear. The intro hints at trying to improve semantic and environment generalization or performance on long-horizon tasks, but does not explain how the method explicitly addresses these challenges. It's only stated that the key-point trajectories "reduce the feature disparity between the vision-language input and action spaces." After reading the full paper, my guess at the motivation is that GR-1 is pre-trained on human video prediction so one could imagine that conditioning on key-points somehow makes better use of the GR-1 initialization than directly predicting actions (since key-points on an image might map more directly to features learned for video prediction). However this reasoning is not explicitly stated anywhere in the paper. Also, the method still provides improvements without GR-1 pre-training so a different explanation/motivation would be needed in that case. Without the video prediction pre-training, I could imagine that the 2D key-point trajectories help with longer horizon behaviors by acting as a plan. If the motivation is planning, a relevant baseline would be to simply perform action chunking [9], i.e predicting multiple actions into the future. One could imagine that action chunking serves a similar purpose to predicting future gripper key-points and is much simpler to implement.

**Novelty**

My larger concern is novelty. The paper is very similar to several papers that predict and condition on gripper key-point trajectories [1, 2, 3, 4, 5, 6]. In particular Any-Point Trajectory Modeling (ATM) [1] and RT-Trajectory [2] are quite similar. These papers are addressed in the related work, but the stated differences don't seem meaningful. The related work says that RT-Trajectory generates linear trajectories rather than particle trajectories, but I'm not sure what that means or why it's a relevant difference. It's also said that ATM predicts trajectories for multiple points in the image rather than just the gripper. However, I'm not sure why predicting only the gripper movement is beneficial (in fact it seems like it could be useful to predict the movements of objects as well). Additionally, tracking just the gripper introduces the additional assumption of a calibrated camera. ATM does not have this assumption. ATM also performs more thorough evaluation by performing experiments on a real robot, and additionally demonstrates improvements from training the point trajectory predictor on human videos and other robot morphologies.

Other relevant papers to discuss in the related work include chain-of-thought with VLAs [7] and visual prompting with VLMs [8, among many others].

**Evaluation**

The evaluation of the proposed method is limited to the simulated CALVIN benchmark, which makes it difficult to determine if the method would generalize to real robot applications. Papers proposing similar methods typically evaluate over a much broader range of simulated and real robot tasks (see cited papers below).

**Presentation/Writing**

Along with the lack of clear motivation in the intro as mentioned above, there are several other issues with the presentation:

- The intro doesn't state that the method predicts 2D key-point trajectories so it's easy for the reader to assume the method is doing video prediction.
- It's not clear whether the video prediction loss is used when fine-tuning GR-1.
- It's not clear from the method section whether the method uses just the architecture from GR-1 or the pre-trained model itself (though it becomes clear in the experiments).
- I would replace the term "diffusion trajectory" with "key-point trajectory" since diffusion describes the generation process and not the type of trajectory.
- The inference (evaluation time) procedure is never described in the method section. I'm assuming that at evaluation time the diffusion trajectory model is used to generate a key-point trajectory which is then used to condition the policy, whereas during training the ground truth key-point trajectories are used to train the policy.
- In section 4.4, the subsection title "Diffusion Trajectory Prompt" should be called "Re-planning" or something similar. This section should also cite model predictive control or receding horizon control work since it seems like that's the technique that is being used here.
- In section 4.4, I could not understand what was meant by "Strategic Prompting"

[1] https://arxiv.org/abs/2401.00025

[2] https://arxiv.org/abs/2311.01977

*[3] https://arxiv.org/abs/2407.15208

[4] https://arxiv.org/abs/2405.01527

[5] https://arxiv.org/abs/2401.11439

[6] https://arxiv.org/abs/2308.15975

*[7] https://arxiv.org/abs/2407.08693

[8] https://arxiv.org/abs/2403.03174

[9] https://arxiv.org/abs/2304.13705

*these papers came out after the contemporaneous work deadline but including them to help with revisions

**Questions:**

- What are the differences between this work and ATM and RT-trajectory?
- Is the video prediction loss used when fine-tuning GR-1?
- What is "strategic prompting" (section 4.4)?

---

> ### Author Response · Authors · 2024-11-25
> **Response (1/2)**
>
> Dear reviewer:
>
> Thank you for your detailed review and thoughtful comments. We truly appreciate your recognition of our work as a valuable contribution to improving long-horizon language-conditioned imitation learning. Additionally, we are grateful for your positive feedback on the experimental setup of our ablation study. We have revised the paper in accordance with your suggestions and will address your remaining questions below.
>
> >The motivation of the method is not entirely clear. [...]
>
> Thank you for pointing that out. We greatly appreciate your observation that our paper does not explicitly explain how the method addresses challenges in long-horizon tasks. Indeed, our perspective is that 2D trajectories can be more directly mapped from an image, and we also believe that 2D trajectory points are easier to map to the final action space. This forms the basis for our statement about “reducing the feature disparity between the vision-language input and action spaces.” We accept your suggestion and will further refine the introduction to clarify this point in the paper.
> As for the similarity to the ACT method, Reviewer v2oo raised a similar concern. To address this, we have conducted a detailed comparison between our method and ACT, which we hope will resolve your concerns.
> We introduced MT-ACT, an action chunk policy designed for multi-task settings, as our baseline.
>
> | **Method**           | **Experiment** | **1**   | **2**   | **3**   | **4**   | **5**   | **Ave. Len.** |
> |-----------------------|----------------|---------|---------|---------|---------|---------|---------------|
> | MT-ACT[1]               | D→D           | 0.884   | 0.722   | 0.572   | 0.449   | 0.353   | **3.03**          |
> | GR1                  | D→D           | 0.822   | 0.653   | 0.491   | 0.386   | 0.294   | **2.65**          |
> | GR1+act chunk        | D→D           | 0.913   | 0.769   | 0.613   | 0.486   | 0.384   | **3.17**          |
> | DTP               | D→D           | 0.924 | 0.819 | 0.702 | 0.603 | 0.509 | **3.55**   |
> | DTP+act chunk     | D→D           | 0.906   | 0.810   | 0.702   | 0.603   | 0.545   | **3.592**         |
>
> Experimental Analysis: Compared to MT-ACT, our method, DTP, outperforms MT-ACT. However, considering the differences in training methods, we added experiments with GR1+act chunk for a more direct comparison between act chunk and our approach. Observing GR1+act chunk, we found that it indeed performs better than the original GR1 but falls short of the performance achieved by DTP. Additionally, we introduced a DTP+act chunk method, which shows a slightly higher success rate compared to DTP alone.
>
> >Compared to ATM and other methods, DTP currently lacks extensive real robot experiments.
>
> Thank you for bringing this up. Due to certain practical constraints, we were unable to conduct complex real-world experiments earlier. However, over the past few days, we successfully collected some real-world robotic data suitable for training diffusion trajectories. The results, including visualizations of the generated 2D trajectories, are now included in Appendix 4.
>
> <CONTINUED>

---

> ### Author Response · Authors · 2024-11-25
> **Response (2/2)**
>
> >What are the differences between this work and ATM and RT-trajectory?
>
> Thanks for bringing this up. ATM is indeed an excellent work that predicts future trajectories through video input. However, we question the precision of generating labels using such an off-the-shelf tracking model, especially given the large number of sampling points required. Additionally, we believe that this approach has limited generalization ability when generating trajectories in unseen scenarios. This is evidenced by the lack of scene generalization experiments in the paper, as well as its stated limitations, which acknowledge that the videos involve only small domain gaps.
> In contrast, our method predicts the complete end-effector trajectory, and experiments have demonstrated its superior generalization performance. Specifically, in the ABC->D experimental setting, our approach improves the long-horizon tasks success rate in unseen scenarios, showcasing its effectiveness.
> As for RT-Trajectory, it is indeed an excellent work, and its idea has been highly inspiring for us. It demonstrates that providing complete end-effector trajectory points as input can significantly improve the generalization of manipulation tasks. However, the trajectories generated by RT-Trajectory through image generation tend to be relatively coarse.
> In contrast, our diffusion trajectory model produces high-quality trajectories that can be directly utilized in end-to-end model inference. Moreover, our framework is designed to treat 2D trajectories as an additional input modality. In comparison, RT-Trajectory's design relies solely on 2D trajectories and images as inputs, which makes the task's success rate heavily dependent on the quality of the generated 2D trajectories. This distinction highlights the robustness and flexibility of our approach.
> We hope the above explanation addresses the concerns you raised regarding the novelty of our approach in the weaknesses section.
>
> >Is the video prediction loss used when fine-tuning GR-1?
>
> We sincerely apologize for any confusion caused. Yes, the video prediction loss is also included in the computation. As indicated in Fig. 2: Network Architecture, the outputs consist of particle trajectory prediction, action, and video prediction.
>
> >What is "strategic prompting" (section 4.4)?
>
> Thank you for pointing this out. Let us further elaborate on the concept of "strategic prompting." During long-horizon tasks, we observed that if the robotic arm's initial pose—after completing a prior task—is either far from the target of the next task or is an uncommon pose not frequently seen in the training set, the likelihood of failure in the subsequent task increases.
> To address this, we introduce a strategy where a predefined intermediate point, referred to as a strategic prompt, guides the robotic arm to first move closer to the next task’s target. At the same time, the arm adjusts its pose to align with more commonly observed poses from the training set. This intermediate adjustment significantly improves the overall success rate of operations.The entire process is visualized in Fig. 4b). In addition, we would like to emphasize that since these strategic points can be prompted, existing vision large language models could be utilized to generate these strategic points, further enhancing the adaptability and robustness of the approach.

---

> > ### Comment · Reviewer_EuXe · 2024-11-26
> > **Response**
> >
> > Thank you for your reply. Below are my remaining concerns.
> >
> > Thanks for the comparison to action chunking. It would be good to include these results in the paper along with the intuition for why your method outperforms action chunking.
> >
> > While I appreciate the addition of visualizations on real data, the standard set by prior work in this area is to evaluate the full method on a robot.
> >
> > My novelty concerns remain. It's true that ATM uses an off-the-shelf point tracking model to obtain ground truth labels and these may be inaccurate. However, DTP gets around this by introducing the additional assumption of a calibrated camera, so I don't think this difference is relevant. I also don't think it makes sense to say that ATM has more limited generalization without showing a direct comparison. Both ATM and DTP are trained on data that looks very similar to the evaluation setting.
> >
> > Thank you for explaining the strategic prompting. How are these intermediate points generated? Are they hand-selected? If so, it  doesn't seem like this is a scalable approach.

---

> ### Author Response · Authors · 2024-11-26
>
> >My novelty concerns remain. It's true that ATM uses an off-the-shelf point tracking model to obtain ground truth labels and these may be inaccurate. However, DTP gets around this by introducing the additional assumption of a calibrated camera, so I don't think this difference is relevant. I also don't think it makes sense to say that ATM has more limited generalization without showing a direct comparison. Both ATM and DTP are trained on data that looks very similar to the evaluation setting.
>
> Thank you for your reply. First, we acknowledge that ATM is an excellent piece of work, and we also reviewed it as concurrent work while developing our approach. Our perspective is that both ATM and Track2Act belong to point tracking-guided methods, where some points in the environment remain stationary. Our intuition, however, is that human operators generally focus more on the end-effector's trajectory estimation, which differs from the point-based tracking methods. While there are similarities with ATM, we believe there are notable distinctions. As mentioned in Track2Act, ATM's policy relies on per-step image observations for tracking prediction. In contrast, our method directly generates the full trajectory from the current frame’s image and language description, without needing to predict a new 2D trajectory at each step.
>
> >Thank you for explaining the strategic prompting. How are these intermediate points generated? Are they hand-selected? If so, it doesn't seem like this is a scalable approach.
>
> Yes, in our current experiments, we use a heuristic approach to directly identify optimal points. For the generated points, we believe that there are many existing 2D object detection methods available. Therefore, the most straightforward approach would be to use these detected points as our strategic prompt input. Additionally, we found that interacting with existing Vision-Language Models (VLMs) also allows us to obtain reasonable strategic prompt points. We have included some preliminary attempts with this approach in Appendix.5.
>
> >1.Thanks for the comparison to action chunking. It would be good to include these results in the paper along with the intuition for why your method outperforms action chunking.
> >2.Presentation/Writing
>
> Thank you very much for your feedback. We have revised the main body of the paper based on your suggestions and hope that these changes will address some of your concerns.

---

### Official Review · Reviewer_v2oo · 2024-11-07

**Soundness:** 3
**Presentation:** 3
**Contribution:** 2
**Rating:** 5
**Confidence:** 4

**Summary:**

This paper proposes an imitation learning method that uses a diffusion trajectory model to guide policy learning. The diffusion trajectory model is first pre-trained on robot videos to map the robot's motion trajectory into the camera image. The policy is then trained with guidance from the output of the trajectory model. Experiments in CALVIN show that this approach can help the policy learn long-horizon tasks better than prior work.

**Strengths:**

The paper is well-written, and the results on the CALVIN benchmark seems to be compelling compared to the included baselines. There are also several ablations that provide insights.

**Weaknesses:**

1. I’m not entirely convinced why the proposed method is better than the end-to-end diffusion policy with action chunking [1, 2]. Predicting action chunks seems similar to predicting the trajectory (or gripper motion), so why is the proposed two-stage approach preferable to training a single policy to predict the long-term actions? Would it be possible to compare to these apporaches?

2. The paper conducted its evaluation on the CALVIN benchmark, but it seems the proposed method underperforms compared to SOTA methods (e.g., RoboUniView [3], MDT [4]). How does the proposed method compare to these works? Is there any reason why they were not included?

3. To train the DTM, the dataset requires the robot's global states to map the trajectory. This assumption limits the method's ability to leverage additional data, such as human / non-robot videos. It would be more interesting to explore a more general way to obtain 2D trajectories more easily, without relying on proprioceptive information.


---
Reference

[1] Chi et al., Diffusion Policy: Visuomotor Policy Learning via Action Diffusion, 2023.

[2] Zhao et al., ALOHA Unleashed: A Simple Recipe for Robot Dexterity, 2024.

[3] Liu et al., RoboUniView: Visual-Language Model with Unified View Representation for Robotic Manipulation, 2024.

[4] Reuss et al., Multimodal Diffusion Transformer: Learning Versatile Behavior from Multimodal Goals, 2024.

**Questions:**

1. In the 10%ABCD->D task, is the Diffusion Trajectory Model also trained on the limited 10% dataset?

2. Is there a specific reason why the wrist camera is necessary? How would DTP perform if only a static camera were used?

---

> ### Author Response · Authors · 2024-11-25
> **Weaknesses Problem Reply**
>
> Dear reviewer:
> Thank you for your detailed review and thoughtful comments. We are pleased to know that you find our DTP compelling in comparison to the included baselines and that you confirm the quality of our ablation study. We have revised the paper following your suggestions and will address your remaining questions below.
> >I’m not entirely convinced why the proposed method is better than the end-to-end diffusion policy with action chunking [1, 2]. Predicting action chunks seems similar to predicting the trajectory (or gripper motion), so why is the proposed two-stage approach preferable to training a single policy to predict the long-term actions? Would it be possible to compare to these apporaches?
>
> We completely agree that incorporating action chunks could further improve performance. We apologize for any confusion this may have caused.
> We would like to clarify our approach: Our task is a language instruction-following task. Our approach aims to improve the robot arm's trajectory by generating diffusion-based trajectories, which better guide the robot arm to perform the correct action in long-horizon tasks. Therefore, while action chunks could further enhance performance, they do not conflict with our method.
> According to your question, we design a expeiment to show our insight. Our generated action head is not a diffusion policy. To enable a fair comparison between action chunks and our method, we introduced MT-ACT, an action chunk policy designed for multi-task settings, as our baseline.
>
> | **Method**           | **Experiment** | **1**   | **2**   | **3**   | **4**   | **5**   | **Ave. Len.** |
> |-----------------------|----------------|---------|---------|---------|---------|---------|---------------|
> | MT-ACT[1]               | D→D           | 0.884   | 0.722   | 0.572   | 0.449   | 0.353   | **3.03**          |
> | GR1                  | D→D           | 0.822   | 0.653   | 0.491   | 0.386   | 0.294   | **2.65**          |
> | GR1+act chunk        | D→D           | 0.913   | 0.769   | 0.613   | 0.486   | 0.384   | **3.17**          |
> | DTP               | D→D           | 0.924 | 0.819 | 0.702 | 0.603 | 0.509 | **3.55**   |
> | DTP+act chunk     | D→D           | 0.906   | 0.810   | 0.702   | 0.603   | 0.545   | **3.592**         |
>
> Experimental Analysis: Compared to MT-ACT, our method, DTP, outperforms MT-ACT. However, considering the differences in training methods, we added experiments with GR1+act chunk for a more direct comparison between act chunk and our approach. Observing GR1+act chunk, we found that it indeed performs better than the original GR1 but falls short of the performance achieved by DTP. Additionally, we introduced a DTP+act chunk method, which shows a slightly higher success rate compared to DTP alone.
>
> >The paper conducted its evaluation on the CALVIN benchmark, but it seems the proposed method underperforms compared to SOTA methods (e.g., RoboUniView [3], MDT [4]). How does the proposed method compare to these works? Is there any reason why they were not included?
>
> Thanks for bringing this up. Regarding RoboUniview, we recognize it as an excellent contemporaneous work in the field. However, it uses depth as an input modality, whereas our setup exclusively uses RGB, making the baseline comparison inconsistent. As for MDT, it is also a notable work. We did not include it in our comparison table because we found that while it performs well in scenarios where training and testing settings are consistent, such as D->D or ABCD->D, its performance declines in unseen scenarios like ABC->D. Specifically, the authors report an average length of 2.5 for ABC->D (https://github.com/intuitive-robots/mdt_policy/issues/9), whereas our model achieves 3.43. Hence, this also demonstrates that our method has better generalization compared to MDT. Additionally, we have incorporated the reviewer’s suggestion and reported MDT's results in our paper.
>
> >To train the DTM, the dataset requires the robot's global states to map the trajectory. This assumption limits the method's ability to leverage additional data, such as human / non-robot videos. It would be more interesting to explore a more general way to obtain 2D trajectories more easily, without relying on proprioceptive information.
>
> Thanks for pointing it out. This is indeed an important consideration. We mentioned this concern in Section 5 (Conclusion and Future Work), where we share a similar perspective: our aim is to obtain diffusion trajectory labels directly, without relying on the camera’s intrinsic and extrinsic parameters.
> In this work, we seek to demonstrate that 2D trajectories can enhance performance in long-horizon manipulation tasks, and our results support this. Additionally, we provide a framework to leverage these generated trajectories effectively.
>
>
> ### References
> [1] Homanga et al.,RoboAgent: Generalization and Efficiency in Robot Manipulation via Semantic Augmentations and Action Chunking. 2024

---

> ### Author Response · Authors · 2024-11-25
> **Adding Questions Reply**
>
> >In the 10%ABCD->D task, is the Diffusion Trajectory Model also trained on the limited 10% dataset?
>
> We apologize for any confusion. The answer is yes, as shown in Table 2 (Ablation Study). The second-to-last row indicates that the diffusion trajectory model (DTM) was trained on a limited 10% dataset.  I’d also like to point out the last row, where '100%✓' means the DTM was trained on the full 100% ABCD->D dataset. As we can see, this leads to performance improvements. Therefore, one potential direction for future work is to pre-train the DTM on larger datasets, which could further enhance its generalization capabilities.
>
> >Is there a specific reason why the wrist camera is necessary? How would DTP perform if only a static camera were used?
>
> Thank you for raising this question. We believe the wrist camera is essential. As we know, the End-Effector outputs actions in 7 degrees of freedom (DOF) — specifically, x, y, z, rx, ry, rz, and gripper state — which require precise predictions along the Z axis.The wrist camera provides the policy with information about the Z direction. Based on this, we consider the wrist camera necessary. If experimental constraints prevent its installation, a multi-view camera setup could serve as an alternative, as we must capture Z-axis information for optimal model performance. The following table answers the question regarding DTP training with only a static camera. Observing the results in the table, it also supports that a wrist camera is valuable.
> | **Method**           | **Experiment** | **1**   | **2**   | **3**   | **4**   | **5**   | **Ave. Len.** |
> |-----------------------|----------------|---------|---------|---------|---------|---------|---------------|
> | DTP w/o wrist camera        | D→D           | 0.736   | 0.523   | 0.358   | 0.254   | 0.176   |**2.05**          |
> | DTP               | D→D           | 0.924 | 0.819 | 0.702 | 0.603 | 0.509 | **3.55**   |

---

> ### Comment · Reviewer_v2oo · 2024-11-28
>
> I thank the authors for their response. The comparison to action chunking resolved my concern. However, after reading the other reviewers' comments, I agree with Reviewer EuXe's point regarding the similarity to the ATM and RT-trajectory lines of work and the lack of novelty. Given that there are no comparisons to these methods, it is difficult to be convinced that the proposed method is a strong approach.

---

> ### Author Response · Authors · 2024-11-29
>
> Thank you for your response. We are happy to hear that our newly added experiments have addressed your concerns. Regarding your reference to Reviewer EuXe's point, we believe you have already reviewed our earlier responses about the differences between our approach and ATM as well as RT-Trajectory. Here, we would like to further emphasize one key distinction: Our method leverages 2D end-effector trajectories as guidance for policy learning, distinguishing it from ATM. In contrast, RT-Trajectory relies on manually provided 2D trajectories, that is human effort costing, especially for long-horizon tasks during deploying the policy.
>
> In fact, when training with the end-effector trajectories, we benefit from our data collection approach, which makes it straightforward to apply data augmentation strategies, such as flipping, translation, and mirroring, directly on 2D images. This enhances the generalization capability of our first training stage. In contrast, ATM relies on extracting trajectories from key points between video frames, which makes it less straightforward to achieve the same flexibility in data augmentation and label expansion.
>
> Improving the generalization in the first stage is critical, as shown in Table 2 (Ablation Study), where we demonstrate that using more training data(100%) in the first stage improves the final success rate of the robot policy (training on 10%). Therefore, we believe our method is not only easier to scale but also experimentally proven to produce better results in the first stage, which directly contributes to higher success rates overall.

---

### Meta-Review · Area_Chair_D6go · 2024-12-19

**Metareview:**

This paper proposes a trajectory-based diffusion policy for long-horizon robot manipulation tasks. While the approach addresses a relevant and challenging problem in robotics, there are significant concerns from reviewers that need to be considered.

Strengths:

- The paper targets an important area in robotics, long-horizon imitation learning, and presents a reasonable framework based on point trajectories.

- Results on the CALVIN benchmark show promising improvements over several baselines.

Weaknesses:

- There is a consensus among reviewers that the proposed method has limited novelty due to its similarity to existing works like ATM and RT-Trajectory. The claimed distinctions are either minor insufficiently substantiated.

- Comparison to several key baseline methods is missing, leaving the significance of the proposed contributions insufficiently justified.

- The evaluation is restricted to the CALVIN benchmark in simulation, which raises concerns about the method’s real-world applicability. Although some real-world trajectory visualizations were added, they are insufficient to address this limitation.

- Reviewers highlighted issues with clarity, including ambiguous motivations, undefined inference procedures, and inconsistent terminology.

Despite the interesting problem and promising results in simulation, the paper falls short in novelty, real-world evaluation, and comparative analysis. These weaknesses collectively outweigh the strengths. Therefore, I recommend rejection of this paper.

**Additional Comments On Reviewer Discussion:**

During the discussion and rebuttal period, reviewers raised concerns about the novelty of the proposed approach, its similarity to prior works like ATM and RT-Trajectory, and the lack of real-world results. While the authors provided additional results, the reviewers believe these additions were insufficient to fully address concerns about novelty and evaluation scope. Ultimately, all reviewers leaned toward rejection, suggesting that the work requires broader evaluations and clearer novelty contributions for future submission.

---

### Decision · Program_Chairs · 2025-01-22

Reject